# Motor Behavioral Deficits in the Cuprizone Model: Validity of the Rotarod Test Paradigm

**DOI:** 10.3390/ijms231911342

**Published:** 2022-09-26

**Authors:** Concordia Lubrich, Paula Giesler, Markus Kipp

**Affiliations:** Institute of Anatomy, Rostock University Medical Center, 18057 Rostock, Germany

**Keywords:** cuprizone, deficit, multiple sclerosis, progression, remyelination, demyelination, gait, motor

## Abstract

Multiple Sclerosis (MS) is a neuroinflammatory disorder, which is histopathologically characterized by multifocal inflammatory demyelinating lesions affecting both the central nervous system’s white and grey matter. Especially during the progressive phases of the disease, immunomodulatory treatment strategies lose their effectiveness. To develop novel progressive MS treatment options, pre-clinical animal models are indispensable. Among the various different models, the cuprizone de- and remyelination model is frequently used. While most studies determine tissue damage and repair at the histological and ultrastructural level, functional readouts are less commonly applied. Among the various overt functional deficits, gait and coordination abnormalities are commonly observed in MS patients. Motor behavior is mediated by a complex neural network that originates in the cortex and terminates in the skeletal muscles. Several methods exist to determine gait abnormalities in small rodents, including the rotarod testing paradigm. In this review article, we provide an overview of the validity and characteristics of the rotarod test in cuprizone-intoxicated mice.

## 1. Introduction

Multiple sclerosis (MS) is a complex, inflammatory, and chronic neurological disorder of the brain and spinal cord. During the initial phase of the disease, inflammation, mediated via the adaptive immune system, clinically results in specific behavioral deficits from which the affected patients can recover either entirely or partially. Depending on the neuroanatomical location of the lesion(s), various behavioral domains can be affected, including the motor-behavioral, visual, or cognitive domains. This disease phase is called “relapsing–remitting MS (RRMS)”. As the disease progresses, the frequency of the development of these relapses decreases. Instead, there is a progressive accumulation of behavioral deficits from which the patients usually do not recover. This secondary disease phase, named “secondary progressive MS (SPMS)”, is driven by a diffuse and chronic inflammatory process inside and around the brain and spinal cord parenchyma [1,2]. Since the disease’s clinical manifestations and pathological consequences vary dramatically among patients, the severity of the disease course is hard to predict on an individual patient level. However, new biomarkers are currently being tested [3]. Eventually, patients initially present with a progressive disease course, which is called “primary progressive MS (PPMS)”.

Focal, inflammatory lesions are the characteristic pathological finding in RRMS, whereas diffuse inflammation and tissue injury are typical for SPMS and PPMS. At the microscopic level, such focal active lesions, which are most frequently found in the white matter, are usually centered on a vessel. The vessel is surrounded by peripheral immune cells (primarily monocytes and lymphocytes), which accumulate within an enlarged perivascular space. In more advanced lesions, these peripheral immune cells invade the surrounding central nervous system (CNS) parenchyma leading to oligodendrocyte degeneration, myelin loss (id est demyelination), microglia, and astrocyte activation, and, eventually, axonal injury. During progressive MS (SPMS and PPMS), focal lesions are less frequent, but diffuse injury, affecting both the white and the grey matter, especially the cortex, dominates the histopathological picture [2,4]. Notably, recent findings suggest that microglia and astrocytes have central roles during the progressive disease process [5,6]. Despite developing highly effective treatments for RRMS, limited progress has been made for progressive MS, especially PPMS [7,8,9]. Current efforts to develop novel progressive MS treatment strategies focus on direct neuroprotective effects and/or the induction of remyelination. While neuroprotective drugs might directly protect axons, synapses, or even entire nerve cells against irreversible damage, the beneficial effect of remyelination induction is believed to be indirect: Among other mechanisms, remyelination strongly reduces neuronal energy consumption and provides trophic axonal support from glia cells [10].

## 2. Preclinical MS Models

Different cell types are involved in regulating and orchestrating MS disease development and progression, making novel drug discovery programs challenging. Preclinical animal models of MS provide the necessary test bed for evaluating the effects of novel therapeutic strategies. Many different models are available, all with their pros and cons. On the one side are the various forms of experimental autoimmune encephalomyelitis (EAE). In these models, the disease is induced by the subcutaneous administration of a CNS antigen dissolved in Freund’s adjuvant supplemented with the heat-killed Mycobacterium tuberculosis strain H37Rv [11]. To further boost the immune system, animals are injected with pertussis toxin into the intraabdominal cavity. Entirely synthetic mycobacterial adjuvants are currently under development [12]. Immunization of mice usually results in ascending paralysis, first affecting the tail, later the hindlimbs, and finally the forelimbs. Depending on the peptide used for immunization and the immunized mouse strain, the clinical course can be either progressive or relapsing–remitting. For example, the immunization of C57BL6 mice with MOG_35–55_ peptide results in a progressive disease course, whereas immunization of SJL-mice with PLP_139–151_ peptide results in a relapsing–remitting disease course [13]. Strategies to evaluate disease severity in EAE-induced mice vary from lab to lab but, in general, are comparable. For example, a score of 1 is equivalent to a limp tail, whereas the hindlimbs are not (yet) affected. A score of 3 is assigned if there is a limp tail and complete paralysis of the hind legs. In even more severe cases, there is complete hind and partial front limb paralysis, a severe condition where euthanasia is strongly recommended. Although several other behavioral domains are affected in the diverse EAE models, including the sensory domain [14,15,16] and cognition [17,18,19], or there is anxiety-like behavior [20,21], it is the motor behavioral deficit that serves as the primary outcome measurement in most of the EAE studies.

Another group of frequently used animal models for drug development is toxin-mediated demyelination models, which allow studying the progress of de- and remyelination and the effects of novel drug compounds on these de- and regenerative processes. In the lysophosphatidylcholine (LPC)-induced demyelination model, LPC is injected into the brain parenchyma where it rapidly integrates into cellular membranes, and increases cell membrane permeability [22,23], leading to myelin sheath disintegration, and subsequently focal demyelination in a matter of hours to days [24,25,26]. Splitting of the intraperiod lines and subsequent breakdown of the entire myelin sheath has been observed at the ultrastructural level [24]. In this model, remyelination occurs in weeks [27,28]. Another commonly-used demyelination–remyelination model is the cuprizone model [29]. In this model, rodents, usually C57BL/6 mice, are intoxicated per os with 0.2–0.3% cuprizone provided either in pellet form or mixed into ground rodent chow. Through a not yet fully understood mechanism, cuprizone intoxication induces oligodendrocyte degeneration resulting in multifocal demyelination of various white and grey matter brain areas. Both apoptosis and ferroptosis have been shown to mediate oligodendrocyte degeneration [30,31], the latter representing a relatively newly described form of iron-mediated cell death. At the mechanistic level, an integrated stress response activation appears to be involved, as recently demonstrated by our group [32]. It should be noted that this model represents various aspects of progressive rather than relapsing–remitting MS [33].

In both models, the LPC model and the cuprizone model, demyelination is followed by a spontaneous regenerative process called remyelination. Both models are, therefore, attractive to study the potency of novel compounds to stimulate (or repress) remyelination. In the case of the cuprizone model, endogenous remyelination is impaired after prolonged cuprizone intoxication, thus allowing the study of the effect of compounds in the non-supportive microenvironment [34,35].

In most cases, histological and ultrastructural readouts are performed to evaluate the effectiveness of novel treatment strategies. Luxol-fast blue stains and immunohistochemistry using antibodies directed against major myelin proteins such as anti-proteolipid protein (PLP) or anti-myelin basic protein (MBP), for example, are frequently applied to estimate the myelination levels of the region of interest [36,37,38]. Ultrastructural studies using electron microscopy can verify the histological results and analyze the axonal myelin diameter ratio, also called “G-ratio”, which is believed to be a sensitive marker for remyelinated axons [39].

Despite the strength of histological analyses to estimate myelin and oligodendrocyte densities and visualize axonal injury or glia activation, it is well known that behavioral impairments and structural deficits do not always correspond to each other in MS and its animal models. While, for example, it has been shown in one study that corticospinal tract lesion volume positively correlates with disability in MS patients [40], other studies have found only weak relationships between radiological findings and clinical consequences [41,42]. Comparably, the frequency and extent of forebrain inflammation do not predict the severity of motor behavioral deficits in the Cup/EAE model [43,44]. Therefore, besides histological and ultrastructural methods, readouts providing information about functional impairment and recovery are attractive means to study compound effectiveness during preclinical trials.

In principle, different functional readouts are available in the different MS animal models. For example, in cuprizone-intoxicated mice, impaired cognition and altered anxiety levels have been described [45,46,47,48]. In the EAE model, Pollak et al. showed that disease induction led to decreased food intake, reduced social exploration, and decreased preference for sucrose solution, which was evident even before the manifestation of motor impairment [49]. While it is beyond the scope of this review article to list all the available functional tests which can be performed in the different MS models, the analysis of motor impairment is one of the most frequently used approaches. As already outlined above, the assessment of motor behavioral deficits is the most commonly applied readout to study therapeutic effectiveness in the different EAE models. By contrast, despite multifocal demyelination, overt motor behavioral deficits are sparse in the cuprizone model, mainly because the spinal cord is unaffected [50]. Therefore, more sensitive analytical approaches have to be applied to this model to estimate the extent of the cuprizone-induced motor impairment [51,52]. In the following section, we will briefly outline different methods to evaluate motor behavior in small rodents. We will then focus on the so-called rotarod performance test and critically reflect on its validity in the cuprizone animal model of MS.

## 3. General Aspects of the Assessment of Motor Behavior in Rodents

Motor behavior is mediated by a complex neural network that originates in the cortex and terminates in the skeletal muscles. Various neuronal centers, such as the association cortex, sensorimotor cortex, subcortical nuclei, cerebellum, or brainstem, intensively communicate to orchestrate the movement of multiple body parts as required to accomplish intended actions, such as walking. There are numerous testing paradigms available to study motor behavior in mice and rats, including the pole test [53,54,55], the beam walking test [53,56,57], the wire hang test [58,59], the cylinder test, the open field test, different running wheels, different swimming test paradigms, the rotarod test, or footprint analyses [60]. These tests have pros and cons, and it is essential to note that there are very few purely motor behavioral tasks. For example, during the cylinder test, the rodents are placed in a transparent cylinder, and the use of their paws during rearing is recorded. Usually, the ratio of left and right forepaw use is determined. Rodents with unilateral 6-OHDA lesions, a model of Parkinson’s disease, exhibit a decline in the use of the contralateral paw when rearing to explore the novel cylinder environment [61]. Although this test is easy to carry out, is relatively fast to conduct, and requires no training of the experimental animal, it is only helpful in detecting the effects of unilateral motor impairment but is less suitable for measuring bilaterally symmetric motor dysfunctions. As another example, during the open field test, which measures the motor behavior of rodents in an open arena, the motor component of the test results is typically confounded by anxiety levels in the animals.

Notably, specific motor behavioral tests are part of the classical EAE scoring outlined above. For example, the simplest way to perform motor tests, in general, involves the grid walking tests, where animals are placed on a metal or wire grid and observed for both forelimb and hindlimb slips. The very same sign is part of most EAE evaluations: When a mouse is dropped on a wire rack, such hindlimb slips are one criterion for monitoring the progression from score 1 to a score of 2 in EAE [62].

Motor coordination in rodents has traditionally been assessed by the rotarod test, which was first described by Dunham and Miya [63], and in those days called “the rolling roller apparatus”. In several studies, the authors have used this apparatus to quantify motor performance in mice treated with different agents, such as chlorpromazine or hydroxyzine [64,65]. The rolling roller apparatus was further developed into an “accelerator” by Jones and Roberts, allowing the rotation speed to increase constantly over a given period. Indeed, this testing paradigm is nowadays one of the most commonly used motor function tests in mice.

The rotarod apparatus consists of a rotating rod that increases rotational speed in a predetermined time interval (see Figure 1). The mouse (or rat, but we will mainly focus on mice in this review article) is placed on the rotating rod, positioned at a height that discourages the experimental animals from jumping off. The base of the apparatus is usually padded for a gentle fall. The primary outcome measure is the latency of the mouse to fall off. Usually, the device is equipped with a timer that automatically stops when the experimental animal falls off. In this way, several animals can be analyzed in parallel. Mice that experienced, for example, experimental stroke showed a reduced latency to fall compared to sham-operated mice [66]. In other words, the time the stroke mice could stay on the rotating rod was lower compared to sham-operated mice. Similarly, subarachnoid hemorrhage mice showed reduced latency before falling compared to sham controls [67]. Furthermore, one might compare the speed at which the experimental animals fall off the rotating rod. Injured animals will fall at a lower speed compared to control animals. However, this test has certain limitations. For example, animals may cling to the beam and rotate with it rather than balancing on the rotating rod. Moreover, some animals refuse to balance and fall as soon as they are placed on the rod. Nevertheless, the rotarod apparatus is the most frequently applied method for testing motor performance in neurosciences using mice or rats as model systems.

In the next section, we will briefly address the cuprizone-induced pathology specifically focusing on motor-related brain regions involved.

## 4. The Cuprizone-Induced Histopathological Changes

As this article mentioned at the beginning, cuprizone intoxication results in primary oligodendrocyte degeneration followed by microglia and astrocyte activation, leading to multifocal demyelination. Furthermore, vulnerable white matter tracts display extensive acute axonal damage. While the demyelinating lesions occur at multiple CNS sides (id est multifocal model), demyelination is almost complete in some regions. By contrast, other CNS areas are less prone to showing total myelin loss in response to the cuprizone insult.

Among the various areas involved in regulating motor behavior, cuprizone induces significant demyelination in the motor cortex, lateral parts of the caudoputamen, parts of the globus pallidus, or the cerebellar nuclei. Sensorimotor coordination requires orchestrated network activity in the brain, mediated by inter- and intra-hemispheric interactions. Various studies have demonstrated that the corpus callosum shows severe demyelination in the cuprizone model. However, as illustrated in Figure 2, the corpus callosum is not uniformly vulnerable. At the anterior commissure level, the corpus callosum’s lateral parts are demyelinated, whereas the medial parts are less severely and reproducibly affected. By contrast, at the rostral hippocampus level, demyelination of the corpus callosum is most severe at its medial parts, whereas the lateral parts are spared. Notably, seminal primate [68] and split-brain patient studies [69,70,71,72] have convincingly demonstrated that the corpus callosum is vital for bimanual motor behavior. In line with these findings, brain imaging studies generally have, although not univocally, revealed that higher fractional anisotropy values, predominantly in the corpus callosum, are predictive of better upper limb control [73,74,75,76,77,78]. Furthermore, physical activity per se can result in structural and/or functional remodeling of callosal fibers. For example, after 1 week of physical exercise training, diffusion tensor imaging (DTI) studies showed an increase in fractional anisotropy and a reduction in radial diffusivity of the corpus callosum, both indicative of structural and/or functional adaptive processes [79].

Other important areas which are pivotally involved in the modulation or propagation of motor behavior, such as the spinal cord white and grey matter or the internal capsule, are not or are only moderately affected in the cuprizone model [50], explaining, as outlined above, the absence of overt motor behavioral deficits in this model.

## 5. Value of the Rotarod Apparatus in Measuring Motor Deficits in the Cuprizone Model

Up to July 2022, 1114 manuscripts were listed in PubMed for the search term “cuprizone”. When “cuprizone AND motor“ was used as the search term, 119 publications were listed. Several different methods were used in these manuscripts to assess motor impairment in cuprizone-intoxicated mice. Vega-Riquer and colleagues, for example, used the horizontal bar test [81], which consists of a brass bar that is fixed ~50 cm above the bench surface. The mouse is placed at the central point of the horizontal bar in a way that only the forepaws grasp the bar. The time the rodents remain on the bar is recorded [82]. Since the ability of a mouse to grip the bar is inversely proportional to the bar diameter, different bar diameters have been used during this test. After 8 weeks of cuprizone intoxication, the authors observed a decline in the horizontal bar test performance, with a further decline during a 30-day remyelination period. Another group showed a decline in motor performance after 5 weeks of cuprizone intoxication using the MotoRater system, a semi-automated system for rodent kinematic gait analysis [83]. Liebetanz and colleagues used a so-called “Motor skill sequence (MOSS)”. This experimental approach is designed to analyze voluntary wheel-running behavior on a sequence of two different types of running wheels: a training wheel with regularly spaced crossbars and a complex wheel with irregularly spaced crossbars [51]. The latter wheel design was chosen to detect subtle changes in motor learning that might depend on bihemispheric integration. Similar customized running wheels were used to display motor deficits in mice after experimental callosotomy [84]. Liebetanz et al. observed, using this complex wheel, a decline in running wheel performance after acute cuprizone-induced demyelination and partial recovery after endogenous partial remyelination [51]. Similar results were obtained by Sullivan and colleagues using a comparable complex running wheel after chronic cuprizone-induced demyelination [85]. Despite these various methods, the rotarod apparatus is the most frequently used method to test for motor performance abnormalities in the cuprizone model.

Table 1 lists the studies conducted so far using the rotarod apparatus to test for motor impairment in cuprizone-intoxicated mice. As demonstrated, various experimental protocols have been used to induce multifocal CNS demyelination with cuprizone concentrations ranging from 0.2–0.7%, and intoxication periods ranging from 1 to 12 weeks. While most studies used the parameter “latency to fall” as the primary outcome measurement, some studies applied additional or alternative outcomes such as the number of animals that can balance the rotating rod for a defined duration [86], or the number of falls (from the rotating rod) and flips (when the animal hangs onto the rod) during the experimental run [87]. Furthermore, while parts of the studies used the accelerating version of the test, others monitored motor performance at a constant speed [88,89]. Notably, the inclusion of a single or several training sessions before determining the motor performance on the rotarod apparatus was another inconsistent variable when comparing the outcomes of the different studies.

Using the accelerating rotarod protocol, the results reported are inconsistent. Some studies did not find any significant difference between the rotarod performance of control and cuprizone-intoxicated mice. Bölcskei and colleagues, for example, intoxicated 8- to 9-week-old mice with 0.2% cuprizone up to 6 weeks. In these mice, the accelerated rotarod protocol was applied with a rotation speed of 4 rpm and acceleration to 40 rpm within 5 min [90]. Mice were trained to run on the wheel for three days. The measurements were performed weekly, and the latency to fall in three separate trials was averaged and statistically compared on each occasion. Although the authors verified demyelination of the corpus callosum utilizing histological stains, ultrastructural studies, and T2-weighted magnetic resonance imaging (MRI), no significant difference was observed between control and cuprizone intoxicated mice at any time point analyzed. In addition, the authors observed no changes in noxious mechanosensation and performance in the grip strength test. The only behavioral parameter found to be different between control and cuprizone-intoxicated mice was a significantly increased rearing behavior in the open field test starting from week 2 till the end of the experiment, indicating increased exploratory behavior and/or a state of impulsivity. Similar results using a similar experimental setting were reported by Polyák and colleagues [91]. Chang and colleagues also reported the absence of a decreased latency using the accelerating rotarod protocol [92]. Notably, in their studies, analyses were performed after 16 days of cuprizone intoxication, when demyelination is incomplete.

In contrast to these ‘negative’ outcomes, Ye and colleagues found a significantly decreased latency after a 5-week cuprizone intoxication period [93]. Notably, therapeutic intervention to ameliorate the cuprizone-induced oligodendrocyte pathology showed promising effects at the histological level (LFB/PAS stain, myelin stains, and ultrastructural studies), whereas no difference was observed concerning the latency to fall. Comparable to the findings of the former study, Liu and colleagues also found a decreased latency at week 5 [94], which was restored by a ketogenic diet used in this trial as a therapeutic intervention. A decrease in latency was also reported by Madadi and colleagues when motor performance was analyzed after chronic cuprizone-induced demyelination. Again, therapeutic intervention, in this case with L-a-aminoadipate, a glutamate homolog working as an astrotoxin for the ablation of astrocytes, rescued the motor performance deficits [95]. Barati and colleagues observed similar effects [96].

Other studies did not apply the accelerating protocol but analyzed the motor performance at a constant rotational speed. For example, Han and colleagues evaluated mice before and after 6 weeks of cuprizone intoxication on the rotarod three times a day for two consecutive days, with rotational speed constantly set at 16 revolutions per minute (rpm). Mice were allowed to run for a maximum of 60 s. If the animal fell, the chronometer was stopped, the animal was put back onto the cylinder, and then the timing was continued. The trial was repeated after 5–10 min. The falls and flips (when the animal hangs onto the cylinder and continues all the way around) were recorded within the 60 s of each trial. The authors made two critical observations. Firstly, the number of falls decreased during the study in control mice (from 10 falls to less than 4 falls), suggesting that the mice either become habituated to the motor task and/or motor learning occurs. Secondly, demyelinated mice showed significantly higher numbers of falls compared to the control mice [101]. Comparably, Yamamoto and colleagues found that after 5 weeks of cuprizone intoxication the number of falls increased, whereas the latency decreased when mice were tested at a constant speed of 28 rpm [87]. Templeton and colleagues observed similar effects [113].

In summary, there is essential evidence from various studies and groups that cuprizone-induced demyelination results in a decline in motor performance, as determined by the rotarod testing paradigm. However, the most consistent results were obtained using a constant rotational speed, whereas the application of the accelerating protocol gave inconsistent results. A study by Buitrago and colleagues is remarkable in this context [115]. When animals, in this case, rats, were repetitively placed on a rotating rod, the test performance improved within the sessions (intrasession) and between the sessions (intersession). Both intrasession and intersession improvement was most remarkable at the beginning but reached a plateau later during the study. Retesting after 8 days of pause revealed retained performance. Notably, rats, while learning to balance on the rotating rod, altered their running technique as the training progressed.

It has recently been shown that motor learning improves recovery from demyelinating injury via enhanced remyelination from new and surviving oligodendrocytes [116]. To what extent repetitive measurements using the accelerating rotarod induce myelin repair, and might therefore interfere with the drug effects, remains to be clarified. Alternatively, it might be that the accelerating rotarod protocol is a priori too demanding to reliably show differences between control animals and experimental animals. Future studies are needed to address these critical aspects.

Finally, we will discuss whether a beneficial therapeutic intervention results in superior motor performance. For obvious reasons, only those studies demonstrating impaired motor performance after cuprizone-induced demyelination can be included in this part.

Some studies have indeed demonstrated that a pharmacological intervention ameliorates a cuprizone-induced decline in rotarod performance. Elbaz and colleagues showed, using a constant speed of 25 rpm and performing three training sessions, a significant reduction in the duration for which mice stayed on the rotating rod in the vehicle, but not Linagliptin-treated mice [89]. Comparably, Hashimoto and colleagues showed that Baicalein attenuates both cuprizone-induced demyelination and motor performance decline [104]. Comparably, Iwasa and colleagues reported that AL-8810 treatment significantly attenuated the cuprizone-induced impairment of motor performance at a rotational speed of 16 rpm [105]. Similar results were obtained when the accelerating protocol was used [94,95,107]. However, some studies were not able to correlate protective effects at the histological level with attenuation in the cuprizone-induced impairment of motor performance [93].

## 6. Summary and Conclusions

To conclude, there is substantial evidence that motor performance is impaired in cuprizone-intoxicated mice and that this motor impairment can be quantified using the rotarod paradigm. However, only some of the studies investigated to what extent cuprizone-induced motor impairment recovers during the endogenous remyelination phase, which even follows acute cuprizone-induced demyelination. One study showed that remyelination is paralleled by functional recovery [95,101]. Other studies either did show that this is not the case [99] or that functional impairment becomes even more severe during the remyelination period [97,98]. The latter results are remarkable as it has been demonstrated that after completed remyelination, axonal degeneration progresses at a low level, accumulating over time [117]. Notably, as some studies did not evaluate motor performance at the end of the cuprizone intoxication period [95,102], no conclusions can be drawn from these studies as to whether remyelination was paralleled by any functional recovery.

Further comparative studies are needed to test which experimental setting can most reliably detect motor deficits after cuprizone-induced demyelination and recovery during myelin repair using the rotarod apparatus. This would, however, require that the reporting of the experimental setup follows standardized guidelines. In our opinion, the following parameters should be included: diameter and texture of the rotating rod, the height of the rod above the bench, number and duration of training sessions, results obtained during the training sessions, number of trials per testing day, as well as a rest period in between the individual trials.

Furthermore, one might consider more complex gait analyses to be used as an indicator of motor performance in this and other MS animal models. In the past, gait analyses have been achieved using the so-called “footprint test”, in which the paws are coated with ink, and the experimental animal is allowed to walk over a sheet of white paper, thus generating a footprint pattern. These were subsequently analyzed for different gait metrics such as stride length, hind-base width, or front-base width [118]. Nowadays, digital video recording systems are available, allowing recording movies of rodents as they walk on a transparent surface, followed by a frame-by-frame analysis. Assessing dynamic parameters of locomotion, such as the step cycle duration, is now possible. Although self-made options have been published [119], three commercially available platforms, CatWalk™ (Noldus Inc., Wageningen, The Netherlands), DigiGait™ (Mouse Specifics Inc., Framingham, MA, USA), and TreadScan™ (CleverSys Inc., Reston, VA, USA), are currently used to perform gait analyses in rodents. The DigiGait™ and TreadScan™ systems are equipped with a motorized treadmill and a high-speed camera that generates digital paw prints from the animal as it runs. In contrast, the CatWalk™ system is equipped with a non-motorized, stable track that allows voluntary gait. The latter system uses the Illuminated Footprints™ technology to record footprints. The great advantage of this type of paw print detection is that there is no risk of falsely detecting parts of the feet (or of the rest of the body, such as the base of the tail) as part of the paw print. The validity of these platforms for determining cuprizone-induced gait deficits remains to be clarified.

## Figures and Tables

**Figure 1 ijms-23-11342-f001:**
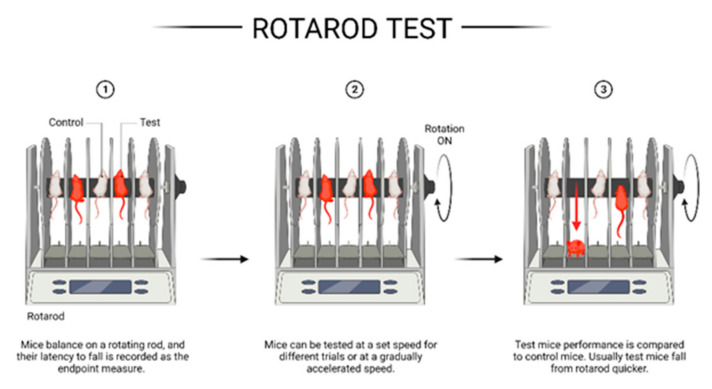
Schematic illustration of the rotarod setup and test principle. Created with BioRender.com (accessed on 14 August 2022).

**Figure 2 ijms-23-11342-f002:**
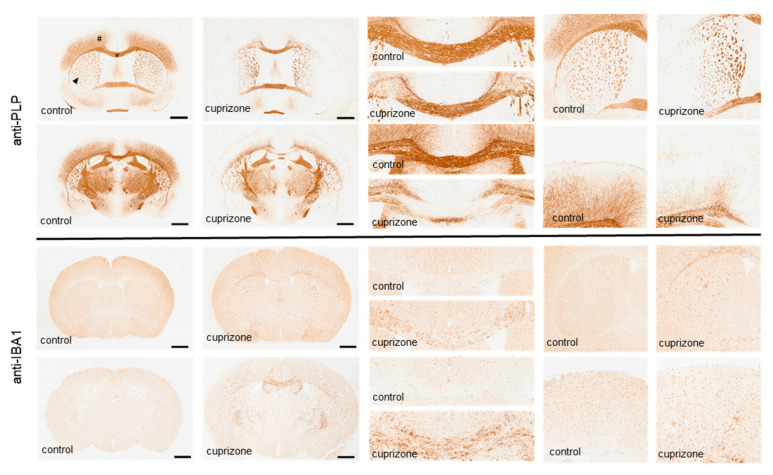
Representative images of anti-PLP and anti-IBA1 stained sections from control and 5 weeks cuprizone-intoxicated mice to demonstrate demyelination and the concomitant activation of microglia. * = midline of the corpus callosum; # = motor cortex; arrowhead = caudoputamen, Scale bar: 1 mm. Adopted from [80].

**Table 1 ijms-23-11342-t001:** Summary of behavioral studies in the cuprizone model using the rotarod testing paradigm. Column 1 (Cit): Citation of the study; Column 2 (Duration): Duration of cuprizone intoxication; Column 3 (Dose): Applied cuprizone dose; Column 4 (Weight): Weight of the mice at the beginning of the cuprizone intoxication; Column 5 (Age): Age of the animals at the beginning of the cuprizone intoxication; Column 6 (Confirmed): Indication whether or not demyelination was histologically confirmed; Column 7 (Timepoint(s)): Timepoints at which rotarod testing was performed. “+” indicates the duration of the remyelination period. For example, week 6 + 3 indicates that 3 wks of remyelination were allowed after a 6-wk cuprizone intoxication period; Column 8 (Setup): Applied experimental setup of the rotarod test; Column 9 (Readout): Parameter used to evaluate motor performance using the rotarod testing paradigm; Column 10 (Training): Indication whether or not a training session or multiple training session were performed prior to the analysis; Column 11 (Main outcome): Main findings of the study. Rpm (revolutions per minute); wks (weeks).

Cit	Duration	Dose	Weight	Age	Confirmed	Timepoint(s)	Setup	Readout	Training	Main Outcome
[97]	4 wks	0.6%	16–18 g	8-wks	no	week 4, week 4 + 2	21 rpm	latency	no	decreased latency
[98]	3–6 wks	0.2%	not given	8-wks	yes	week 3–6, week 6 + 6	16, 24, or 32 rpm	number of falls	no	increased number of falls
[99]	6 wks	0.2%	not given	8-wks	no	week 6, week 6 + 3	4 to 40 rpm in 5 min	latency	no	no change
[100]	6 wks	0.2%	not given	8-wks	yes	week 6 + 11 days	30 rpm, max 200 s	latency	no	decreased latency
[101]	6 wks	0.2%	not given	7-wks	yes	week 6 + 2, 4, 6	15–16 rpm, max 60 s	number of falls	no	increased number of falls
[91]	5 wks	0.2%	20–25 g	8-wks	yes	week 3, 4, and 5, week 5 + 3, and 5 + 4	5 to 40 rpm in 300 s	latency	yes	no change
[95]	12 wks	0.2%	not given	7-wks	yes	week 12 + 2	4 and 40 rpm in 3 min	latency	yes	decreased latency
[102]	8 wks	0.4%	15–18 g	60 d	no	week 8 + 3	8, 15, 30, and 35 rpm, max 60 s	normalized latency	yes	decreased latency
[96]	12 wks	0.2%	20–25 g	8-wks	yes	week 12 + 2	4 to 35 rpm in 3 min	latency	yes	decreased latency
[103]	8 wks	0.2%	not given	8-wks	yes	week 8	28 rpm, max 300 s	latency	yes	decreased latency
[90]	6 wks	0.2%	not given	8–9 wks	yes	week 1–6	4 to 40 rpm in 5 min	latency	yes	no change
[92]	3 wks	0.4%	17–20 g	6-wks	yes	day 16	4 to 35 rpm in 3 min	latency	yes	no change
[89]	4 wks	0.7–0.2%	20–25 g	6-wks	yes	week 4	25 rpm	latency	yes	decreased latency
[86]	5 wks	0.2%	20 g	8–10 wks	yes	week 5	6 rpm, max 120 s	latency	no	decreased latency
[104]	6 wks	0.2%	not given	10-wks	yes	week 6	20 rpm, max 300 s	latency	yes	decreased latency
[105]	5 wks	0.2%	not given	10-wks	yes	week 5	16 rpm, max 600 s	latency, number of falls	no	decreased latency, increased number of falls
[106]	1 wk	0.2%	not given	8-wks	no	week 1	4 to 40 rpm in 2 min	latency	no	no change
[88]	30 days	6 mg/kg	25–50 g	not given	yes	day 30	25 rpm, max 120 s	latency	yes	decreased latency
[107]	6 wks	0.2%	18–20 g	7–8-wks	yes	days 40, 41, and 42	4 to 40 rpm in 120 s	number of falls	no	increased number of falls
[87]	5 wks	0.2%	not given	10-wks	yes	week 5	28 rpm, max 300 s	latency, number of falls and flips	no	decreased latency, increased number of falls and flips
[108]	10 wks	0.2%	not given	10-wks	yes	week 5 and 10	20 or 28 rpm, max 300 s	latency, number of falls and flips	no	decreased latency, increased number of falls
[93]	5 wks	0.2%	15–17 g	6-wks	yes	week 5	5 to 40 rpm	latency	yes	decreased latency
[109]	5 wks	0.2%	not given	8–10-wks	yes	week 5	32 rpm, max 300 s	latency, number of falls	yes	decreased latency, increased number of falls
[110]	48 days	0.2%	18–22 g	8–9-wks	yes	between day 34 and 48	28 rpm, max 300 s	latency, number of falls	yes	decreased latency, increased number of falls
[111]	45 days	0.2%	18–20 g	7-wks	yes	week 5	30 rpm, max 300 s	latency, number of falls	yes	decreased latency, increased number of falls
[94]	5 wks	0.2%	not given	7-wks	yes	week 5	4 to 40 rpm	latency	no	decreased latency
[112]	6 wks	0.2%	22 g	6-wks	yes	week 6	4.5 m/min	latency	yes	no change
[113]	6 wks	0.3%	not given	6–8-wks	yes	week 6	28 rpm, max 120 s	latency	no	decreased latency
[55]	6 wks	0.2%	20–25 g	8-wks	yes	week 6	25 rpm, max 300 s	latency	no	decreased latency
[114]	5 wks	0.2%	not given	10-wks	yes	week 5	28 rpm, max 300 s	latency, number of falls and flips	no	decreased latency, increased number of falls and flips

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
