# Peer review of "Motor Behavioral Deficits in the Cuprizone Model: Validity of the Rotarod Test Paradigm"

_ijms, 2022, doi:10.3390/ijms231911342_

Round 1

Reviewer 1 Report

The review is devoted to the validity of rotarod test application for multiple sclerosis model (cuprizone treated mice). The manuscript is well written, contains required information in the introduction with definition of what is multiple sclerosis, describes current preclinical animal models of multiple sclerosis, ways to assess motor activities in rodents, peculiarities of cuprizone model, and results of studies of motor activities in this model using rotarod apparatus. Future prospects were given as a conclusion.   

Minor issues:

Line 269-271: The reference should be provided.

line 381-383: The last name of first author is different in the paper cited.

The paper can be accepted provided that above mentioned minor issues will be corrected.

Author Response

Thank you for reviewing carefully our joint manuscript. We have adopted the mansucript according to your two kind suggestions. Your comments helped us to further improve the quality of this work.

Reviewer 2 Report

The cuprizone model is widely used in MS research. It has many advantages, it is easily reproducible, and the histopathological phenomena have been precisely analyzed. It has been a long-standing demand of researchers in the field to be able to monitor the cuprizone effect in vivo. Pet MRI is one of the possible methods for this, however, this device requires a significant financial burden and a serious professional background. Tracking motor performance with the simple rotarod method seems like an obvious solution. Various research groups are engaged in a long scientific debate on this issue. This publication summarizes and properly positions the assessment of motor performance in the application of the cuprizone model.
The authors interestingly raise the question, according to which motor performance is maintained despite remyelination. The question arises whether the cuprizone model - in this case - is perhaps more suitable for examining progressive forms of MS. Regarding this issue, the opinion of the authors may be interesting. In addition, are they planning to accurately validate the rotarod method (e.g. following motor performance in parallel with pet MR).

Author Response

Thank you for reviewing our manuscript and providing valuable input. We now mention in the revised version of the manuscript that the cuprizone model represent various aspects of progressive MS and provide a citation of this claim.

To compare motor-behaviour with different imaging modalities - among PET - is an ongoing project, and we hope that we can publish parts of these results in the near future.